# Time-Dependent Analysis of Plasmalogens in the Hippocampus of an Alzheimer’s Disease Mouse Model: A Role of Ethanolamine Plasmalogen

**DOI:** 10.3390/brainsci11121603

**Published:** 2021-12-02

**Authors:** Abul Kalam Azad, Abdullah Md. Sheikh, Md. Ahsanul Haque, Harumi Osago, Hiromichi Sakai, Abu Zaffar Shibly, Shozo Yano, Makoto Michikawa, Shahdat Hossain, Shatera Tabassum, Garu A., Xiaojing Zhou, Yuchi Zhang, Atsushi Nagai

**Affiliations:** 1Department of Neurology, Faculty of Medicine, Shimane University, Izumo 693-8501, Japan; akazad@med.shimane-u.ac.jp (A.K.A.); shibly@med.shimane-u.ac.jp (A.Z.S.); agaru429@med.shimane-u.ac.jp (G.A.); zhouxj93@med.shimane-u.ac.jp (X.Z.); zhangyuchi1014@icloud.com (Y.Z.); 2Department of Microbiology, Jagannath University, Dhaka 1100, Bangladesh; 3Department of Laboratory Medicine, Faculty of Medicine, Shimane University, Izumo 693-8501, Japan; abdullah@med.shimane-u.ac.jp (A.M.S.); ahsanul@uap-bd.edu (M.A.H.); syano@med.shimane-u.ac.jp (S.Y.); tabassum@med.shimane-u.ac.jp (S.T.); 4Department of Pharmacy, University of Asia Pacific, Dhaka 1205, Bangladesh; 5Department of Biochemistry, Faculty of Medicine, Shimane University, Izumo 693-8501, Japan; biochem1@med.shimane-u.ac.jp; 6Department of Biosignaling and Radioisotope Experiment, Faculty of Medicine, Shimane University, Izumo 693-8501, Japan; hisakai@med.shimane-u.ac.jp; 7Department of Biotechnology and Genetic Engineering, Mawlana Bhashani Science and Technology University, Tangail 1902, Bangladesh; 8Department of Biochemistry, Graduate School of Medical Sciences, Nagoya City University, Nagoya 467-8601, Japan; michi@med.nagoya-cu.ac.jp; 9Department of Biochemistry and Molecular Biology, Jahangirnagar University, Dhaka 1342, Bangladesh; shahdat@juniv.edu

**Keywords:** LC-SRM/MS, plasmalogen, ROS, phagocytosis, GNPAT, glial cells

## Abstract

Plasmalogens are alkenyl-acyl glycerophospholipids and decreased in post-mortem Alzheimer’s disease (AD) brains. The aim of this study is to investigate the time-dependent changes of plasmalogens in the hippocampus of an AD model mouse (J20). Plasmalogen levels at 3, 6, 9, 12 and 15 months were analyzed by liquid-chromatography-targeted-multiplexed-selected-reaction-monitoring-tandem-mass-spectrometry (LC-SRM/MS). Reactive oxygen species (ROS) levels were evaluated using dichlorofluorescein diacetate (DCF-DA). Plasmalogen synthesizing enzyme glycerone-phosphate O-acyltransferase (GNPAT) and late endosome marker Rab7 levels were quantified by Western blotting. GNPAT localization, changes of neuronal and glial cell numbers were evaluated by immunostaining. Compared to wild-type mice (WT), total plasmalogen-ethanolamine, but not plasmalogen-choline levels, were increased at 9 months and subsequently decreased at 15 months in J20 mice. A principal component analysis of plasmalogen-ethanolamine species could separate WT and J20 mice both at 9 and 15 months. Both GNPAT and Rab7 protein were increased in J20 mice at 9 months, whereas GNPAT was decreased at 15 months. ROS levels were increased in J20 mice except for 9 months. Our results suggest that increased plasmalogen-ethanolamine could counteract ROS levels and contribute to the phagocytosis process in J20 mice at 9 months. Such results might indicate a transient protective response of plasmalogen-ethanolamine in AD conditions.

## 1. Introduction

Plasmalogens (Pls) are a subclass of glycerophospholipids, containing fatty alcohol at the sn-1 and fatty acid at the sn-2 positions of their glycerol backbone [1]. Due to the presence of a vinyl ether bond, Pls can act as an antioxidant [2]. Additionally, the content of Pls determines the physiochemical properties, such as the fluidity of the cell membrane, which is important for diverse cellular functions, including ion transport regulation, membrane enzyme activities, receptor-mediated signaling, and membrane fusion during phagocytosis, endocytosis, exocytosis, and synaptic transmission [1,2,3,4,5,6,7]. During the phagocytosis process, Pls-PE facilitates the fusion processes required for sealing the membrane to form the phagosome [8]. Rab7, a late endosomal protein, is required for the formation and maturation of the phagolysosome by fusion between late endosomal vesicles and lysosomal vesicles [9,10], indicating the possibility of an association between Pls-PE and Rab7 change. Importantly, a defect in Pls synthesis or reduced Pls levels are associated with different peroxisomal diseases, neurodegenerative diseases, and metabolic diseases [11,12]. However, the role of Pls level change in the pathology of these diseases remained unknown [1].

Alzheimer’s disease (AD) is a common neurodegenerative disease characterized by extracellular amyloid β (Aβ) peptide deposition as plaques and intracellular neurofibrillary tangles [13]. Several genetic, clinical, animal, and in vitro studies demonstrated the importance of Aβ deposition in AD pathology [14]. Deposited Aβ can cause neurodegeneration directly by interacting with the cell membrane or indirectly by inducing neuroinflammation [15], oxidative stress [16], and peroxisome dysfunction [17]. Plaque-induced reactive oxygen species (ROS) increase peroxisomal dysfunction and decrease peroxisomal Pls synthesizing enzymes, including glycerone-phosphate O-acyltransferase (GNPAT), leading to reduced Pls levels [18,19]. Oxidative stress can degrade Pls by ROS. In AD brains, deposited Aβ peptides are generated from the amyloid precursor protein (APP) by β- and γ-secretase activities. Usually, APP is processed by α-secretases in a non-amyloidogenic pathway. When membrane Pls content is decreased, such physiochemical membrane alteration changes the processing from non-amyloidogenic to the amyloidogenic pathway by increasing β- and γ-secretase activity [5,20]. Notably, Pls levels are reported to decrease in the post-mortem brain samples of AD subjects and AD model aged-mice [19,21,22,23,24]. Such events might induce a vicious circle in which Aβ promotes free radical production and Pls degradation, leading to a favorable condition for β- and γ–secretase activity and production of the peptide. Hence, it is important to understand the changes of Pls during different stages of AD and its relationship with oxidative stress during the progression of the disease.

Several methods, including thin-layer chromatography (TLC), high-performance liquid chromatography (HPLC), or enzymatic measurement methods, have been employed to measure Pls in biological samples [25,26,27,28]. However, these methods cannot separate and analyze the different Pls species, which could be important to understand the disease pathology. Liquid chromatography-tandem mass spectrometry (LC-MS/MS) is a sensitive method to analyze and quantify molecules in biological samples. Recently, we have developed an LC-targeted multiplexed SRM/MS method where sn-1 fatty alcohols and sn-2 fatty acids of individual 22 Pls-PC and 55 Pls-PE species were characterized using a selected reaction monitoring approach. In addition, the individual concentration of Pls species was accurately quantified in biological samples [29]. The method was used in this study to evaluate the time-dependent changes of Pls molecular species in the hippocampal brain tissues of an Alzheimer’s disease model mouse and correlate the changes with ROS and GNPAT levels. Furthermore, the late endosomal marker Rab7 was evaluated to determine the association of Pls change with the phagocytosis process. We found that Pls-PE levels were increased transiently in AD model mice at 9 months of age.

## 2. Materials and Methods

### 2.1. Chemicals and Reagents

The internal standards (ISs) 1-pentadecanoyl-2-oleoyl(d7)-sn-glycero-3-phosphocholine (PC (d7–33:1)) and 1-pentadecanoyl-2-oleoyl(d7)-sn-glycero-3-phosphoethanolamine (PE (d7–33:1)) were purchased from Avanti Polar Lipids (Alabaster, AL, USA). LC-MS grade ammonium formate and ammonium acetate were purchased from Sigma-Aldrich (St. Louis, MO, USA). LC-MS grade methanol (MeOH) and methyl tert-butyl ether (MTBE) were obtained from Wako Pure Chemical Industries (Osaka, Japan).

### 2.2. Animals

Male mice that had the human amyloid precursor protein (hAPP) transgene (J20) were used as an AD model and their wild-type (WT) littermate as a control. This mouse model overexpresses the hAPP gene with two mutations (Swedish-K670N/M671L and Indiana-V717F mutations) linked to familial AD and develops amyloid pathology and behavioral alterations in a progressive fashion [30,31]. Mice were maintained at a maximum of four mice per cage after one month of age in 12 h light/dark cycle (lights on at 7:00 a.m.). Food and water were provided ad libitum until sample collection. J20 and WT groups were identified and verified by PCR using specific primers [32]. Mice were kept in fasting for 6 h before sample collection. The Ethical Committee of Shimane University School of Medicine approved the study (Approval number: IZ29-28). All animal experimental procedures were performed following the guidelines and the regulations for experimentation at Shimane University, Japan.

### 2.3. Phospholipid Extraction

Phospholipids were extracted according to the previously reported lipid extraction method [33] with some modifications. Briefly, 10 mg of weighed hippocampal brain tissue was homogenized in 462 μL ice-cold methanol containing ISs (133 pmol PC (d7–33:1) and 282 pmol PE (d7–33:1)). Samples were centrifuged at 6000 rpm for 2 min at 4 °C, and the supernatant was collected. Then, 1540 μL MTBE was added and incubated overnight with shaking at room temperature (RT). Afterward, 128 μL of 0.15 M ammonium acetate solvent was added to 1/3 of the supernatant and centrifuged at 3000 rpm for 15 min at 4 °C. The upper organic layer was collected, and the bottom layer was re-extracted with 411 μL of solvent mixture (10:3:2.5-MTBE: methanol: 0.15 M ammonium acetate, by volume). Finally, the collected lipid extract was dried under a vacuum evaporator and stored at −80 °C until LC-MS/MS analysis. Before LC-MS/MS analysis, the dried lipid extract was dissolved in 667.3 μL of methanol/10 mM ammonium formate (9:1, by volume).

### 2.4. Liquid Chromatography-Targeted Selected Reaction Monitoring Mass Spectrometry (LC-SRM/MS) Analysis

The phospholipid extracts were analyzed to identify and quantify Pls and alkyl-acyl-phospholipids using the LC-targeted multiplexed SRM/MS method [29]. Briefly, a Shimadzu HPLC system coupled to a triple quadrupole mass spectrometer equipped with an electrospray ionization (ESI) source (Nexera X2, LCMS-8030, Shimadzu Co., Kyoto, Japan) was used. The chromatograph consisted of a DGU-14 AM degasser, an LC-10AD pump, an SIL-HTC autosampler held at 4 °C, and a thermostatted column compartment. The ESI source settings were as follows: heat-block and dissolved-line temperatures were 400 °C and 250 °C, respectively, and nebulizer and drying gas flow rates were 2 and 15 L/min, respectively. Nitrogen (N_2_) gas was used as the collision and nebulizer gases. The mobile phase was 10 mM ammonium formate (solvent A) and 100% methanol (solvent B). A sample was loaded onto a YMC-Triart C18 analytical column (3 μm, 2.0 × 100 mm, YMC co., Kyoto, Japan) with a guard cartridge held at 40 °C. Separation was performed using a binary gradient system at a flow rate of 0.2 mL/min. The gradient program was as follows: 0 min, 80% B; 8 min, 100% B; 15 min, 100% B; 15.1 min, 80% B; and 25 min, 80% B. Data acquisition and analysis were performed using LabSolutions software (version 5.60SP2 software; Shimadzu).

The quantification of Pls-PE, alkyl-acyl-PE, Pls-PC, and alkyl-acyl-PC was performed using a product ion neutral loss of *m*/*z* 141.0 and *m*/*z* 184.0, respectively. The area of each target was corrected by the IS area ratio of sample to standard to eliminate the matrix effect. The quantitative values were calculated using the corrected area with calibration curves of PE (d7–33:1; 711.6 > 570.5) and PC (d7–33:1; 753.6 > 184). The measurement points of PE (d7–33:1) was 5, from 0.00072 to 0.45120 pmol/2 µL injection, and that of PC (d7–33:1) was 6, from 0.00014 to 0.4256 pmol/2 µL injection. The peak areas of all analyte Pls and alkyl-acyl phospholipids species were within the range of the calibration curve. Finally, amounts were expressed as pmol/mg of the tissue sample. Then, the individual concentration of isobaric Pls-PE species was determined based on the relative percentage of these isobaric species. The relative percentage was determined by monitoring selected reaction monitoring transitions targeting product ions *m*/*z* 362.0, 392.0, 390.0, 420.0, and 418.0 for Pls-PE species with 16:0, 18:0, 18:1, 20:0, and 20:1 fatty alcohols at the sn-1 position, respectively [29].

### 2.5. Tissue Preparation for Immunohistochemistry

J20 and WT mice were used for immunohistochemical analysis at 9 months. All the mice (3 mice in each group, *n* = 6) were deeply anesthetized with isoflurane and transcardially perfused with normal saline, followed by 4% paraformaldehyde in 100 mM phosphate-buffered saline (PBS, pH 7.4) for tissue fixation. The mice brains were removed and post-fixed into the same fixative for 24 h at 4 °C. Then, the brain samples were cryoprotected with 30% sucrose in 100 mM PBS for 48 h at 4 °C. The frozen tissues were serially sectioned coronally into tissue blocks of 2 mm thickness, and afterward, 8-µm thickness tissue slices were prepared for staining using a cryostat (Leica, Wetzlar, Germany).

### 2.6. Immunofluorescence Staining

Mice brain tissue sections were incubated in a blocking solution containing 5% goat serum and 0.1% Triton X-100 in PBS for 45 min at RT. Then, the tissue was incubated with primary antibodies anti-Iba1 IgG (rabbit polyclonal, 1:200, Abcam, Cambridge, UK), anti-GNPAT IgG (rabbit polyclonal, 1:400, Abcam, Cambridge, UK), anti-GFAP IgG (rabbit polyclonal, 1:200, Dako, Carpinteria, CA, USA), and anti-NeuN IgG (mouse monoclonal, 1:200, MAB377, Millipore, Billerica, MA, USA) overnight at 4 °C. After washes with PBS, the tissue sections were incubated with species-specific fluorescence-conjugated secondary antibodies (goat anti-rabbit IgG Texas red or goat anti-mouse IgG FITC, 1:200, Santa Cruz, CA, USA) at RT for 1 h. Hoechst 33258 (Sigma, St. Louis, MO, USA) staining (10 µg/mL) was used to identify the nuclei of the cells. After staining, sections were mounted with Ultramount (DAKO) and photographed with a fluorescence microscope (NIKON, E600). The number of immunoreactive cells were counted in 3 consecutive sections 2 mm apart in 5 random fields of images at 40× magnification by using NIH ImageJ analysis software version 1.52 (NIH, Bethesda, MD, USA).

Double immunofluorescence staining was performed to determine the localization of GNPAT with Iba1^+^ and GFAP^+^ immunoreactive cells in J20 mice. Brain tissue slices were incubated in a blocking solution. Then, the tissue slices were incubated with anti-Iba1 IgG (mouse polyclonal, 1:200, Santa Cruz, CA, USA) and anti-GFAP IgG (mouse polyclonal, 1:200, Santa Cruz, CA, USA) overnight at 4 °C. The following day, the tissue was incubated with fluorescence conjugated goat anti-mouse IgG Texas red (1:200, Santa Cruz, CA, USA) at RT for 1 h. Then, the tissue slice was incubated with rabbit anti-GNPAT IgG overnight at 4 °C. The next day, the tissue section was incubated with fluorescence conjugated goat anti-rabbit IgG FITC (1:200, Santa Cruz) at RT for 1 h. Hoechst 33258 (Sigma, St. Louis, MO, USA) staining (10 µg/mL) was used to identify nuclei of the cells. After staining, sections were mounted with Ultramount (DAKO) and photographed with a fluorescence microscope (NIKON, E600).

### 2.7. Western Blotting Analysis

Hippocampal tissues both from J20 and WT were collected and homogenized in 20X wt/vol of ice-cold RIPA lysis buffer (1X phosphate buffer saline, pH 7.4, 1% Nonidet p-40, 0.5% sodium deoxycholate, 0.1% SDS, 10 μg/mL PMSF, 10 μg/mL aprotinin). Under the reducing condition, 60 μg total protein of the brain tissue was separated by 10% sodium dodecyl sulfate-polyacrylamide gel electrophoresis (SDS-PAGE) and transferred to PVDF membranes (Millipore, Billerica, MA, USA). After blocking 1 h at RT, the membranes were incubated with primary antibodies anti-GNPAT IgG (rabbit polyclonal, 1:1000, Abcam, Cambridge, UK), anti-Rab7 IgG (1:2000, mouse polyclonal, Abcam, Cambridge, UK) and anti-β-actin IgG (1:1000, mouse polyclonal, Santa Cruz, CA, USA) at 4 °C overnight. Subsequently, the membranes were incubated with an infra-red (IR) dye-conjugated anti-rabbit or anti-mouse IgG antibody (1:5000; LI-COR Bioscience, Lincoln, NE, USA) for 9 month samples and a horseradish peroxidase-conjugated anti-rabbit (1:5000, MilliporeSigma, MA, USA) or anti-mouse IgG antibody (1:5000, Santa Cruz, CA, USA) for 1 h at RT. Blots for 6, 12, and 15 month samples were developed using a chemiluminescent system (GE Healthcare, Amersham, UK). Immunoreactions were visualized using an Odyssey infrared imaging system (LI-COR Bioscience) and AMERSHAM Image Quant 800 detection system (GE Healthcare, Amersham, UK). The band intensity was quantified by densitometry and the NIH ImageJ analysis software version 1.52 (NIH, Bethesda, MD, USA). Individual expression levels of GNPAT or Rab7 were normalized to the expression levels of β-actin.

### 2.8. Reactive Oxygen Species (ROS) Level Measurement

Reactive oxygen species (ROS) levels were determined with some modification following the previously reported method [34]. Briefly, 100 µL of freshly prepared brain tissue homogenates (4 mice in each group, *n* = 40) were diluted in 1.4 mL of 100 mM PBS (pH 7.4). Next, 15 µL of 500 µM dichlorofluorescein diacetate (DCF-DA) in methanol were added and incubated with mild shaking for 30 min at 37 °C. After centrifugation at 11,800 *g* for 20 min at 4 °C, the pellet was collected. The pellet was vortexed at 0 °C in 1.4 mL of 100 mM PBS (pH 7.4) and again incubated with mild shaking for 60 min at 37 °C. Fluorescence was measured with a Hitachi 850 spectrofluorometer at wavelengths of 488 nm for excitation and 525 nm for emission. The cuvette holder was maintained at 37 °C. ROS was quantified from the dichlorofluorescein standard curve in methanol.

### 2.9. Statistical Analyses

Integrated data were transferred to Microsoft Excel for an individual species concentration analysis. Statistical significance was assessed using the Student’s *t*-test (paired, two-tailed). After applying autoscaling, a supervised orthogonal partial least square discriminant analysis (OPLS-DA) [35] and an unsupervised principal component analysis (PCA) [36] were performed using online MetaboAnalyst 5.0 software (www.metaboanalyst.ca (accessed on 6 September 2021)). In the PCA, a score plot of the first and second principal components was generated. All quantitative data are presented as mean ± SD (standard deviation). A two-tailed value of *p* < 0.05 was considered statistically significant. All figures were prepared with Microsoft Excel.

## 3. Results

### 3.1. Total Pls-PE, Alkyl-Acyl-PE, Pls-PC, and Alkyl-Acyl-PC in the Hippocampus of J20 and WT Mice

The time-dependent changes of total Pls-PE, alkyl-acyl-PE, Pls-PC, and alkyl-acyl-PC were analyzed by the LC-SRM/MS method in both J20 and WT mice brain hippocampus tissue. Compared to WT, total Pls-PE levels were increased at 9 months and subsequently decreased at 15 months in J20 mice (Figure 1A and Appendix A). Alkyl-acyl-PE was increased at 3 and 9 months (Figure 1B and Appendix A). Pls-PC levels were not changed at any time points (Figure 1C and Appendix A). Total alkyl-acyl-PC was increased significantly at 15 months in J20 mice compared to WT (Figure 1D and Appendix A).

Next, multivariate analyses, including OPLS-DA and PCA, were done to check the distribution pattern of Pls-PE, alkyl-acyl-PE, Pls-PC, and alkyl-acyl-PC metabolism in the J20 and WT littermate mice. For Pls-PE species, the OPLS-DA score plot clearly separated the J20 and WT mice groups at all time points. However, PCA analysis could separate J20 and WT mice only at 9 and 15 months. The first principal component effectively and distinctly separated the J20 and WT groups at 9 and 15 months (Figure 2). For Pls-PC and alkyl-acyl-PC, the OPLS-DA score plot separated J20 and WT mice groups. In the PCA analysis, the first principal component could not separate effectively at any time point (Appendix A).

### 3.2. Molecular Species of Pls-PE, Alkyl-Acyl-PE, Pls-PC, and Alkyl-Acyl-PC in the Hippocampus of J20 and WT Mice

Next, the time-dependent change of Pls-PE and alkyl-acyl-PE species (Appendix A), and Pls-PC and alkyl-acyl-PC species (Appendix A) were analyzed. Pls-PE species including PE (P-16:0/22:5), PE (P-16:0/22:4), PE (P-18:0/20:4), PE (P-18:0/20:3), PE (P-18:0/22:6), PE (P-18:0/22:5), PE (P-18:1/20:2), and PE (P-18:1/22:2) were increased at 9 months in J20 mice compared to WT. At 15 months, PE (P-18:0/16:0), PE (P-18:0/18:1), PE (P-18:1/18:1), PE (P-18:0/18:0), PE (P-18:0/20:3), PE (P-18:1/20:2), PE (P-18:1/20:1), and PE (P-18:0/22:4) were decreased at 15 months in J20 mice (Appendix A). Five Pls-PE, Pls-PC, and alkyl-acyl-PC species with the highest loading scores in the PCA analysis are listed in Appendix A. We found that PE (P-16:0/22:3) and PE (P-18:1/20:2) contributed to separating J20 and WT mice at 9 and 15 months, and increased at 9 months but decreased at 15 months.

### 3.3. GNPAT Protein Expression and its Localization in the Hippocampus of J20 and WT Mice

Our results showed that Pls-PE metabolism was changed at 9 and 15 months in J20 mice compared to age-matched WT littermates. Next, Pls-synthesizing key enzyme, GNPAT, protein levels in the hippocampus were evaluated at 6, 9, 12, and 15 months of J20 and WT mice by Western blotting and quantified by densitometry. Densitometric analysis showed that GNPAT expression was increased at 9 months and subsequently decreased at 15 months in J20 mice compared to WT. At 6 and 12 months, the protein levels were similar in J20 and WT mice (Figure 3A,B). We further checked the immunoreactivity of GNPAT at 9 months. Immunophotomicrograph showed that GNPAT was positive in the neuronal cells in CA1 and other regions of the hippocampus (Figure 3C).

Double immunofluorescence staining of GNPAT with Iba1^+^ (microglia marker) or GFAP^+^ (astrocyte marker) were done to identify whether glial cells were positive for GNPAT. The immunophotomicrograph showed that both microglia and astrocyte were also positive for GNPAT in J20 mice (Figure 4A). Then, the changes of neurons (NeuN^+^), astrocyte, and microglial cell numbers at 9 months were evaluated. Our cell counting result showed that GFAP^+^ and Iba1^+^ positive cell numbers were increased (Figure 4D–G), whereas neuronal cell number was decreased in J20 mice at 9 months (Figure 4B,C).

### 3.4. Change of Rab7 Protein at the 9 Months in the Hippocampus of J20 and WT Mice

It has been reported that both Pls-PE and Rab7 are required for phagosome formation in the phagocytosis process [8,9,10]. Further, the late endosome marker, Rab7, was evaluated at 9 months by Western blotting. The densitometric analysis showed that Rab7 protein expression was increased at 9 months in J20 mice compared to WT (Figure 5A,B).

### 3.5. Oxidative Stress Level Changes in the Hippocampus of J20 and WT Mice

We measured reactive oxygen species (ROS) levels in the hippocampus tissue at 3, 6, 9, 12, and 15 months of J20 and WT mice using DCF-DA as a probe. The results showed that ROS levels were increased at 3, 6, 12, and 15 months in J20 mice compared to WT, whereas no change was found at 9 months in J20 mice (Figure 5C).

## 4. Discussion

This study reports that at 9 months, the transient increase of Pls-PE might develop a protective response against Aβ pathology. By performing a time-dependent analysis of the Pls profile, GNPAT expression, and ROS level change, we found that high GNPAT expression elevated Pls-PE levels, which might mitigate Aβ-induced ROS. In addition, the increase of the late endosomal marker Rab7 also coincided with this Pls-PE change, suggesting a functional association of Pls-PE with another protective response, such as the phagocytosis process. These results indicate a way for understanding the protective role of Pls-PE in AD conditions.

Studies in serum, plasma, and cerebrospinal fluid suggested that Pls levels could differentiate between mild cognitive impairment (MCI) and AD patients, as well as AD severity [37,38,39,40]. However, three studies found Pls-PE deficiency [22,41,42] and three studies found no change of Pls-PE levels but a decreased level of Pls-PC in the brain tissue of post-mortem AD patients [23,24,43]. For the AD model mouse study, Pls-PE levels were reduced at the late stage of Aβ plaque pathology [19,21]. This time-course study covering early, progressive, and late stages of AD pathology also found Pls-PE deficiency at 15 months. A previous study demonstrated that Pls-PE deficiency depends on Aβ deposition, and Pls-PE reduced in AD model aged mice [21]. Aβ plaque deposition first appeared at 9 months and was extensively deposited at 15 months in the hippocampus area of J20 mice [31,32,44]. Therefore, our result suggested that Pls-PE metabolism might be different at the early and late stages of Aβ plaque deposition. Multivariate analyses, such as PCA and OPLS-DA, also confirmed the change of Pls-PE metabolism at the early and late stages of Aβ plaque deposition in this study.

Peroxisome dysfunction and ROS-mediated degradation are the main reasons of Pls-PE alterations in the AD brain [1,12]. Aβ plaque deposition increased ROS and disrupted peroxisomal Pls synthesizing key enzymes expression, resulting in the reduction of Pls-PE levels in the post-mortem AD brain [18,19]. In this study, ROS levels were increased in J20 mice except at 9 months. At the same time, elevated GNPAT expression at 9 months was subsequently decreased at 15 months in J20 mice. High GNPAT expression might have elevated Pls-PE levels at 9 months. Pls-PE is a potent endogenous antioxidant, and PUFA containing Pls-PE supplements inhibited ROS production in the microglial cell [45]. A recent study suggested that Pls ensure the survivability of microglia against long-term cytotoxicity through their antioxidant properties [46]. Therefore, it is tempting to speculate that increased Pls-PE at 9 months might induce a protective response against early Aβ plaque deposition in J20 mice through ROS mitigation. Further studies are needed to determine the molecular mechanism of how PUFA containing Pls-PE mitigates ROS production in AD conditions.

The previous report showed that neurons, astrocytes, and microglia were positive for GNPAT expression, but under stress conditions, change of GNPAT levels were related to glial cells, not neuronal cells [19]. Our study also showed that neuron and glial cells are positive for GNPAT. J20 mice at 9 months exhibited a 32% loss of neurons, a 163% increase in the number of CD68-positive microglia, and a 62% increase in the number of GFAP-positive astrocytes in the CA1 hippocampus area [31]. In line with this report, the number of microglia and astrocytes were increased, and the number of neurons was decreased at 9 months in J20 mice. These results suggested that increased glial cells might contribute to the transient increase of GNPAT and Pls-PE at 9 months in J20 mice.

A previous study demonstrated that PUFA containing Pls-PE plays a vital role in membrane fusion [3] and the endocytosis process [47] essential for phagocytosis. Pls-PE species containing 20:3, 20:4, 22:4, 22:5, and 22:6 FAs at the sn-2 position facilitate the membrane fusion process to form phagosome and regulate phagocytosis process in macrophage cell [8]. Our study showed that Pls-PE species that have PUFAs (22:4, 22:5, 22:6, 20:4, 20:3, 20:2) were increased at 9 months (Appendix A). At the same time, late endosome marker Rab7, necessary for the fusion of late endosomal and lysosomal vesicles to form phagolysosome, was increased in J20 mice. In addition, the Pls-PE species contributed to highest five loadings produced by PC1 and PC2 analysis at 9 months contained 20:4 and 20:3 fatty acids (Appendix A), which are essential for the phagocytosis process [8,48]. Therefore, the Pls-PE increase at 9 months might be involved in the phagocytosis process as a protective response to early Aβ deposition in J20 mice.

On the other hand, total Pls-PE and GNPAT expression levels were decreased at 12 months in J20 mice. Some Pls species were significantly reduced at 12 months in J20 (Appendix A) compared to WT, consistent with a previous study [16]. Aβ plaque deposition was increased continuously in J20 mice. These elevated Aβ plaque deposits mediated oxidative stress might intensify Pls-PE metabolism at the subsequent stages. In favor of this idea, we found that ROS levels were increased extensively, and GNPAT and total Pls-PE levels were decreased at 15 months when Aβ plaque deposition increased remarkably in J20 mice [44].

Previous studies also reported that soluble and oligomer Aβ caused Aβ-induced ROS-mediated toxicity in the subjects of MCI, AD, and AD models [49,50,51]. Our results also showed that ROS levels were increased at 3 and 6 months when total Pls levels were not changed in J20 mice compared to WT. Thus, it can be assumed that many factors, such as Aβ-induced ROS, expression of glial cells, and stage of Aβ plaque deposition, might be involved in the change of Pls in AD conditions.

Our study provides an important indication of the protective role of Pls-PE in AD conditions. However, some limitations of this study should be noted. First, our study showed high GNPAT expression and Pls-PE elevation at the early Aβ plaque stage in AD model mice. Hence, the interaction of early Aβ plaque with glial cells might induce some pathways that result in GNPAT expression and a Pls-PE increase. Identification of such pathways could be important for a better understanding of the effect of Pls-PE change in AD pathology. Second, our study found the functional association of Pls-PE with the phagocytosis process in AD conditions. A previous in vitro study demonstrated the molecular mechanism of how Pls-PE plays an essential role in the lipopolysaccharide-induced phagocytosis process [48]. Future studies will be worthful to find out the molecular pathways explaining the role of Pls-PE in the phagocytosis process in AD conditions. Third, we found that the progressive Aβ plaque deposition intensifies Pls deficiency in the late stages, and previous studies suggested that Pls deficiency also aggravate Aβ plaque deposition pathology by increasing β- and γ-secretase enzyme activity [5,24,52]. In further studies, we plan to check the detailed molecular mechanism of how Pls deficiency plays a role in Aβ plaque deposition in J20 mice.

## 5. Conclusions

In conclusion, our result demonstrated that in J20 mice, Pls-PE changes at early Aβ plaque deposition might be important for counteracting ROS levels and contributing to the phagocytosis process. On the other hand, at the late stage, progressive Aβ deposition decreases Pls-PE levels, which might contribute to the increase of Aβ plaque deposition in J20 mice.

## Figures and Tables

**Figure 1 brainsci-11-01603-f001:**
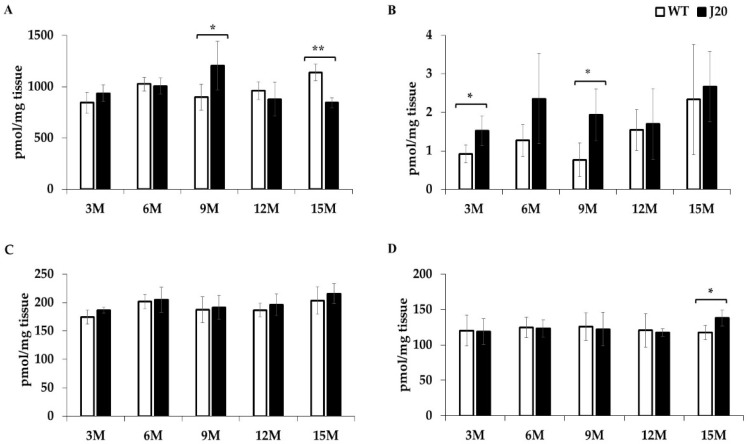
Time-dependent change of total Pls-PE, alkyl-acyl-PE, Pls-PC, and alkyl-acyl-PC in the hippocampus of J20 and WT mice. The concentration of total Pls-PE, alkyl-acyl-PE, Pls-PC, and alkyl-acyl-PC was measured by LC-SRM/MS, as described in the Materials and Methods. The changes of total Pls-PE (**A**), alkyl-acyl-PE (**B**), Pls-PC (**C**), and alkyl-acyl-PC (**D**) were shown. The data are presented here as means ± SD of four mice in a group. Statistical significance is denoted as follows: * *p* < 0.05, ** *p* < 0.01 vs. wild-type mice of the same age. M indicates months.

**Figure 2 brainsci-11-01603-f002:**
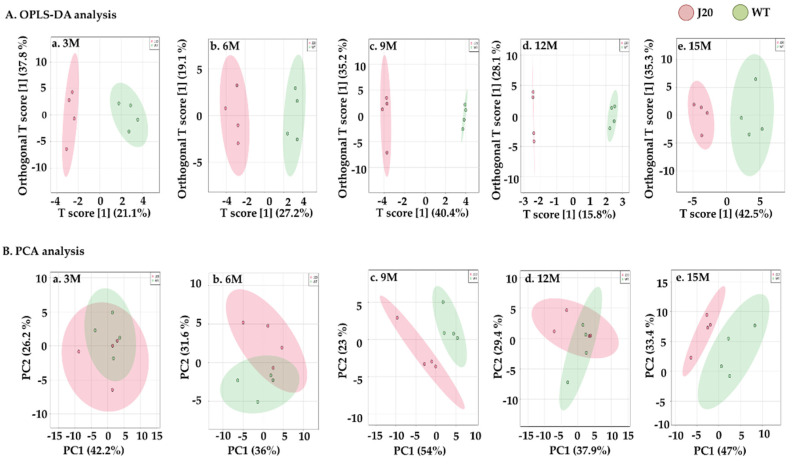
Orthogonal partial least square discriminant analysis (OPLS-DA) and principal component analysis (PCA) of Pls-PE in the J20 and WT mice. Four mice were used in each group for J20 and WT mice at each time point. OPLS-DA analysis (**A**) at 3 M (**A**.**a**), 6 M (**A**.**b**), 9 M (**A**.**c**), 12 M (**A**.**d**), and 15 M (**A**.**e**), and PCA analysis (**B**) at 3 M (**B**.**a**), 6 M (**B**.**b**), 9 M (**B**.**c**), 12 M (**B**.**d**), and 15 M (**B**.**e**) were conducted with the measured Pls-PE by using MetaboAnalyst 5.0 software. Pink-colored circles indicate Pls-PE distribution in J20 mice, and green-colored circles indicate Pls-PE distribution in WT mice. M indicates months.

**Figure 3 brainsci-11-01603-f003:**
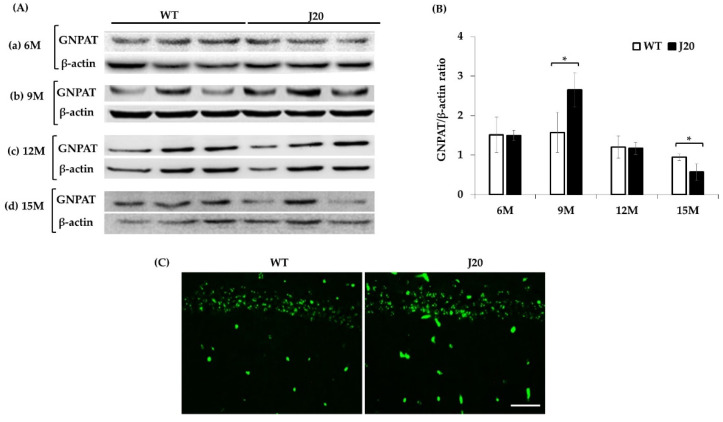
Time-dependent change of GNPAT protein expression in the hippocampus of WT and J20 mice. Total GNPAT expression levels of WT and J20 were evaluated by Western blotting, as described in the Materials and Methods. Representative Western blotting data at the age of 6 M (**A**.**a**), 9 M (**A**.**b**), 12 M (**A**.**c**), and 15 M (**A**.**d**) are shown. Densitometric analyses are shown (**B**). The expression and localization of GNPAT reactivity was checked by immunofluorescence staining at 9 months. Representative photomicrographs of GNPAT protein expression are shown (**C**). The numerical data expressed as average ± SD of 3 mice in a group. Statistical significance is denoted as follows: * *p* < 0.05. Scale bar = 50 µm (**C**). M indicates months.

**Figure 4 brainsci-11-01603-f004:**
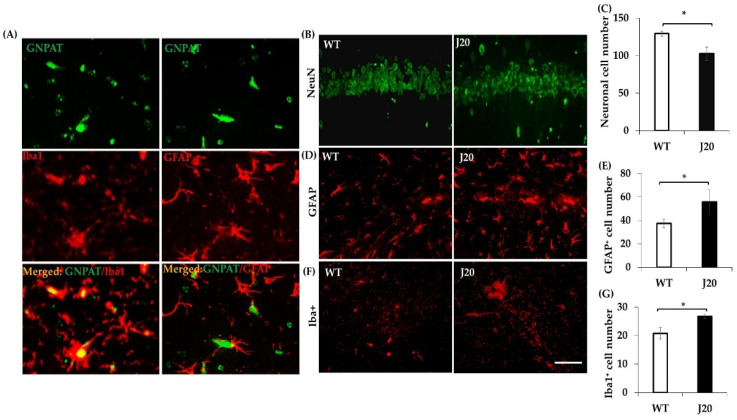
Localization of GNPAT with glial cells in J20 mice, and change of neuronal and glial cell at 9 months in J20 and WT mice. Double immunofluorescence staining of GNPAT with Iba1 and GFAP was performed, as described in the Materials and Methods. Representative immunophotomicrographs are shown (**A**). The change of neuronal and glial cells in the hippocampus of J20 and WT mice were evaluated by NeuN, Iba1, and GFAP immunofluorescence staining. The number of cells were counted in stained tissue sections at 40× magnification. The representative immunophotomicrographs and analysis results of NeuN (**B**,**C**), GFAP (**D**,**E**), and Iba1 (**F**,**G**) are shown. The numerical data is expressed as the average ± SD of 3 mice in a group. Statistical significance is denoted as follows: * *p* < 0.05. Scale bar = 20 µm (**A**) and 100 µm (**B**–**F**).

**Figure 5 brainsci-11-01603-f005:**
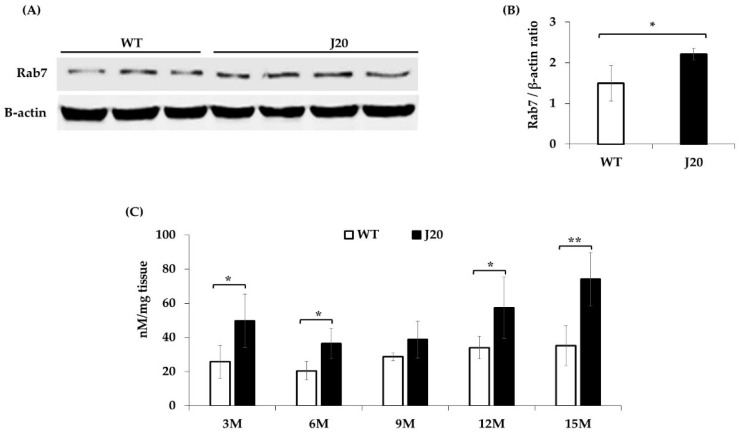
Rab7 expression at 9 months, and time-dependent change of ROS at 3, 6, 9, 12, and 15 months in the hippocampus of WT and J20 mice. Total Rab7 expression levels were analyzed by the Western blotting technique. Representative Western blotting data of J20 (*n* = 4) and WT (*n* = 3) mice are shown (**A**). Densitometric analysis was performed using ImageJ software, and is shown (**B**). ROS levels were quantified by a DCF-DA probe assay. The analyzed result is shown (**C**). Statistical significance is denoted as follows: * *p* < 0.05, ** *p* < 0.01. M indicates months.

## Data Availability

All data of this study are shown in the report.

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
