# Peer review of "Time-Dependent Analysis of Plasmalogens in the Hippocampus of an Alzheimer’s Disease Mouse Model: A Role of Ethanolamine Plasmalogen"

_brainsci, 2021, doi:10.3390/brainsci11121603_

Round 1

Reviewer 1 Report

Here are some of the suggestions/comments that will further improve the the manuscript - 

  1. Figure 2 is not entirely visible in the manuscript and it was hard to determine the exact analyses performed
  2. The AD model used is J20, however, all the legend for almost all the figures compare WT with TG. If I am not wrong, TG stands for 3xTg AD mouse model. Is this a typo or authors compared WT with TG instead of J20?
  3. Did the authors check the PL levels beyond 15 months, if the mice survived beyond 15 months? This reviewer is curious to see if the PL levels drop further beyond 15 months or stay the same.
  4. Authors should explain the significance of estimating the Rab7 levels?

Author Response

Response to reviewer’s comments

We appreciate the reviewer for taking the time to review our manuscript and for the constructive suggestions to improve our paper. The comments are seriously considered, and our point-by-point responses to each comment are described below. We think that those comments have allowed us to significantly improve our manuscript. We hope that we have adequately responded to the concerns raised by the reviewer. In closing, let us thank you once again for your extremely cogent comments that helped us improve the quality of our paper.

Reviewer #1

  1. Figure 2 is not entirely visible in the manuscript and it was hard to determine the exact analyses performed

Response: Thank you for your suggestion. We changed the Figure 2 (P.7 ,ln.15) according to your advice.

  1. The AD model used is J20, however, all the legend for almost all the figures compare WT with TG. If I am not wrong, TG stands for 3xTg AD mouse model. Is this a typo or authors compared WT with TG instead of J20?

Response: Thank you for pointing out these very important aspects. We changed all the legend for all the figures compare WT with TG (J20 relaced TG) according to the reviewer’s suggestion.

  1. Did the authors check the PL levels beyond 15 months, if the mice survived beyond 15 months? This reviewer is curious to see if the PL levels drop further beyond 15 months or stay the same.

Response: Thank you very much for your good suggestion. We are also curious about plasmalogen levels beyond 15 month of age. However, previous studies reported Pls-PE deficiency at late stage (18 months) of AD pathology in aged familial AD model mouse indicating that at aged mice Pls reduction has already been reported. Thus, in this study we aimed to check the Pls metabolism at early, progressive, and late stages of AD pathology (P. 10, ln 28-30).  With respect to the reviewer’s opinion, we are planning to check lyso plasmalogens, plasmalogens, reactive oxygen species (ROS), plasmalogen synthesizing enzymes glycerone phosphate O-acyltransferase (GNPAT) at the ages of 18, 21, 24 and 28 months in J20 and Wild type littermate mice in our further studies.

  1. Authors should explain the significance of estimating the Rab7 levels?

Response: Thank you very much for pointing out these very important aspects. The significance of Rab7 estimation has been narrated in the introduction (P. 2, ln. 23-26), result (P.9, ln. 31, 32) and discussion (P. 11, ln. 12-27),

Reviewer 2 Report

My comments:

  1. Figure 2 should be organized better.
  2. A pathway analysis figure may be great, which shows how plasminogen deficiency could play a role in amyloid deposition.
  3. Is Pls-PE deficiency could also impact other diseases, including neurodegenerative disease? 
  4. Is it possible to measure the levels of plasminogen may be able to measure the severity of AD? Is it possible to distinguish MCI and AD with plasminogen levels?

Author Response

Response to reviewer’s comments

We appreciate the reviewer for taking the time to review our manuscript and for the constructive suggestions to improve our paper. The comments are seriously considered, and our point-by-point responses to each comment are described below. We think that those comments have allowed us to significantly improve our manuscript. We hope that we have adequately responded to the concerns raised by the reviewer. In closing, let us thank you once again for your extremely cogent comments that helped us improve the quality of our paper.

Reviewer #2

  1. Figure 2 should be organized better.

Response: Thank you for your suggestion. We changed the Figure 2 (P.7 ,ln.15) according to your advice.

  1. A pathway analysis figure may be great, which shows how plasminogen deficiency could play a role in amyloid deposition.

Response: Thank you for your good suggestion. The function of plasmalogen species is not well understood like proteins. Therefore, using the lipidomic software, it is very difficult to analyze the pathway of plasmalogens. Since time is limited and no plasmalogen functional pathways are reported so far, it is very difficult for us to make a pathway analysis figure in this study. In this study, we discuss in the discussion section about how plasmalogen deficiency could play a role in amyloid deposition (P.11, ln. 54-55; and P.12, ln.2-4).

  1. Is Pls-PE deficiency could also impact other diseases, including neurodegenerative disease? 

Response: Thank you for pointing out these very important aspects. We included in the introduction section (P.2, ln. 26-29) as follows ‘ Importantly, defect in Pls synthesis or reduced Pls levels are found to be associated with different peroxisomal diseases, neurodegenerative diseases and metabolic diseases. However, the role of Pls level change in the pathology of this disease remained unknown'.

  1. Is it possible to measure the levels of plasminogen may be able to measure the severity of AD? Is it possible to distinguish MCI and AD with plasminogen levels?

Response: Thank you very much for pointing out these very important issues. We added in the discussion section (P. 10, ln. 24-29) as like ‘Studies in serum, plasma, and cerebrospinal fluid suggested that Pls level differentiates between mild cognitive impairment (MCI) and AD as well as AD severity’.

Round 2

Reviewer 2 Report

The authors fulfilled my suggestions. Thank you